

# Certifying quantum separability with adaptive polytopes

**Ties-Albrecht Ohst[1][⋆], Xiao-Dong Yu[2], Otfried Gühne[1] and H. Chau Nguyen[1]**

**1** Naturwissenschaftlich–Technische Fakultät, Universität Siegen,
Walter-Flex-Straße 3, 57068 Siegen, Germany
**2** Department of Physics, Shandong University, Jinan 250100, China

⋆ ties-albrecht.ohst@uni-siegen.de

## Abstract

The concept of entanglement and separability of quantum states is relevant for several fields in physics. Still, there is a lack of effective operational methods to characterise these features. We propose a method to certify quantum separability of two- and multiparticle quantum systems based on an adaptive polytope approximation. This leads to an algorithm which, for practical purposes, conclusively recognises two-particle separability for small and medium-size dimensions. For multiparticle systems, the approach allows to characterise full separability for up to five qubits or three qutrits; in addition, different classes of entanglement can be distinguished. Finally, our methods allow to identify systematically quantum states with interesting entanglement properties, such as maximally robust states which are separable for all bipartitions, but not fully separable.

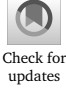

# 1   Introduction

Entanglement is nowadays perceived as one of the hallmarks of quantum mechanics, which not only underlies the foundations of quantum mechanics and quantum technology, but also deeply influences our understanding of physics in various different areas, ranging from condensed matter physics [1,2] to gravity [3–5]. Formally, entangled states are those that cannot be prepared by classical communication and local operations of the parties [6]. Modelling the classical communication by a random variable $\lambda$, this implies that a bipartite entangled state $\rho^{AB}$ cannot be written as a convex combination of states factorised at the parties,

$$\rho^{AB} = \sum_{\lambda} p_{\lambda} \sigma_{\lambda} \otimes \tau_{\lambda}, \tag{1}$$

where $\sigma_{\lambda}$ and $\tau_{\lambda}$ are density operators in Alice's and Bob's spaces, respectively, and $p_{\lambda}$ are probability weights of $\lambda$. States of the form (1) are said to be separable.

    Significant effort has been devoted to methods to determine whether a state is entangled or not [7–13]. As separable states form a convex set, depicted in Fig. 1a, a state can be 'witnessed' to be entangled if one can find a hyperplane that separates it from the set. On the contrary, proving a state to be separable is significantly complicated, requiring testing it against all possible entanglement witnesses or, equivalently, searching over all possible decompositions of the form (1). Accordingly, various methods have been proposed to demonstrate entanglement [7,8], but techniques to verify that a state is separable are scarce. In fact, it has been perceived as a 'notoriously difficult' problem in quantum information theory [14,15]. On the other hand, certification of separability can be essential to optimise quantum information processing protocols, where entanglement between certain parties is difficult to establish and only local operations are available. This has been discussed, for example, in conference key agreement [16], superdense coding [17] and reconstruction of states from marginals [18,19].

    Known approaches to the problem are based on sequences of semidefinite programs that are proven to certify separability for sufficiently high order [20, 21]. Later, iterative algorithms in which states are tried to be transferred into maximally mixed ones with infinitesimal non-separability-breaking transformations have been applied [22, 23]. Further works used a version of Gilbert's algorithm to the convex membership problem [24,25], the method of truncated moment sequences [26] and sets of inequalities in terms of Bloch representations [27].

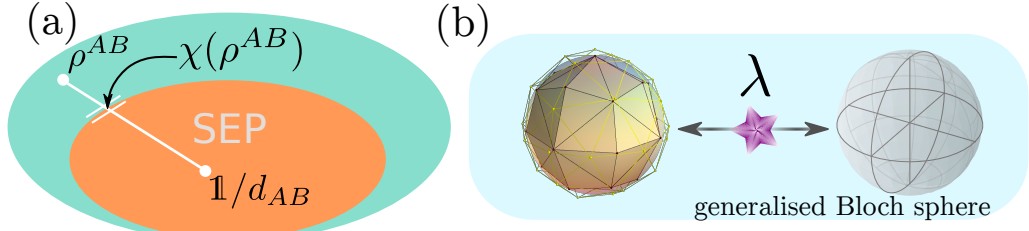

Figure 1: (a) Schematic sketch of the convex set of separable states. The visibility $\chi(\rho^{AB})$ of a quantum state $\rho^{AB}$ corresponds to the fraction of separable states in the convex hull with the maximally mixed state, Eq. (4). Inner and outer polytopes approximating Alice's set of states (generalised Bloch sphere) give rise to lower and upper bounds of $\chi$, respectively. (b) Sketch of the polytope approximation. Alice's generalised Bloch sphere (on the left) is replaced with a polytope while Bob's generalised Bloch sphere (right) is left unchanged. The parameter $\lambda$ indicates the random variable correlating Alice's and Bob's states. Upon approximating Alice's states by a polytope, it takes values as vertices of the polytope.

Recently, neural networks have been used for a parametrisation of separable states to tackle the problem of certifying separability [28] and even variational quantum algorithms for the problem exist [29]. These present methods are highly computationally demanding, therefore applicable only for special states or low-dimensional systems.

Filling this gap, we introduce a method of adaptive polytopes for certifying the separability of quantum states. We show that the resulting algorithm indicates a strong evidence of being nearly optimal independent of the structure of the states in all benchmarks. Being highly efficient, the algorithm is directly applicable to quantum systems of relatively high dimensions and systems with many particles, far beyond results given by the other known methods. In fact, the algorithm not only allows for the certification of a single targeted state, but also for families associated to it, and even for the investigation of different entanglement robustnesses. As an illustration, we apply the technique to explore the geometry of the boundary of the set of separable states for bipartite as well as multiparticle systems.

## 2 Adaptive polytopes for bipartite systems

### 2.1 Polytope approximation

We start with rewriting the set of separable states in Eq. (1), denoted SEP, as the convex hull of product states,

$$\text{SEP} = \text{conv}(\mathcal{S}_A \otimes \mathcal{S}_B), \tag{2}$$

where $\mathcal{S}_A \otimes \mathcal{S}_B = \{\sigma \otimes \tau : \sigma \in \mathcal{S}_A, \tau \in \mathcal{S}_B\}$, with $\mathcal{S}_A$ and $\mathcal{S}_B$ being the sets of states at Alice's side and at Bob's side, which we will call the generalised Bloch spheres. Let $\mathcal{P}$ be a convex subset of Alice's operators with unit trace. Following (2), we consider $\text{conv}(\mathcal{P} \otimes \mathcal{S}_B)$. If we choose an inner polytope $\mathcal{P}_{\text{in}}$ and an outer one $\mathcal{P}_{\text{out}}$ to approximate Alice's generalised Bloch sphere, such that $\mathcal{P}_{\text{in}} \subseteq \mathcal{S}_A \subseteq \mathcal{P}_{\text{out}}$ (see Fig. 1b for an illustration) then it follows that

$$\text{conv}(\mathcal{P}_{\text{in}} \otimes \mathcal{S}_B) \subseteq \text{conv}(\mathcal{S}_A \otimes \mathcal{S}_B) \subseteq \text{conv}(\mathcal{P}_{\text{out}} \otimes \mathcal{S}_B). \tag{3}$$

While the set $\text{SEP} = \text{conv}(\mathcal{S}_A \otimes \mathcal{S}_B)$ cannot be efficiently computed, we show that $\text{conv}(\mathcal{P}_{\text{in/out}} \otimes \mathcal{S}_B)$ can be. In fact, they can be formulated as a standard optimisation of a linear objective function and semidefinite constraints, known as a semidefinite program (SDP), for

which efficient algorithms exist [30]. Indeed, let the polytope $\mathcal{P}$ be described by a set of $N$ vertices, $\mathcal{P} = \{\sigma_\lambda\}_{\lambda=1}^N$, then $\rho^{AB} \in \text{conv}(\mathcal{P} \otimes \mathcal{S}_B)$ means that there exist $N$ positive operators on Bob's space $\{\tilde{\tau}_\lambda\}_{\lambda=1}^N$ such that $\rho^{AB} = \sum_{\lambda=1}^N \sigma_\lambda \otimes \tilde{\tau}_\lambda$ where the $\tilde{\tau}_\lambda$ do not necessarily have trace 1. This can be understood as a minimisation of a constant function with semidefinite constraints, known as a feasibility SDP. More quantitatively, approximating the Bloch sphere by a polytope from inside and outside allows one to directly lower bound and upper bound various types of entanglement robustnesses, see Appendix B. As an illustration, we consider here the white noise mixing threshold to an entangled state such that it becomes separable,

$$\chi(\rho^{AB}) = \max\left\{ t \geq 0 : \rho_t^{AB} \in \text{SEP} \right\} \tag{4}$$

where $\rho_t^{AB} = t\rho^{AB} + (1-t)\mathbb{1}/d_{AB}$ and $d_{AB}$ is the total dimension of the system. By choosing a polytope $\mathcal{P} = \{\sigma_\lambda\}_{\lambda=1}^N$ to approximate $\mathcal{S}_A$, one obtains an SDP to approximate $\chi(\rho^{AB})$ by

$$\chi_{\mathcal{P}}(\rho^{AB}) = \begin{array}{ll} \max & t \\ \text{w.r.t.} & t, \tilde{\tau}_\lambda \succcurlyeq 0 \\ \text{s.t.} & \rho_t^{AB} = \sum_{\lambda=1}^N \sigma_\lambda \otimes \tilde{\tau}_\lambda. \end{array} \tag{5}$$

In practice, if $\mathcal{P}$ approximates the generalised Bloch sphere from inside (outside), one obtains a lower (upper) bound of $\chi(\rho^{AB})$, see also Fig. 1a. Outer polytopes give rise to a useful tool to detect entanglement if one of the parties is a qubit and the dual version of the SDP (5) can then be used to construct tailored entanglement witnesses [31,32], see Appendix A. This is because the simplicity of the geometry the Bloch sphere of the qubit allows for an efficient construction of the outer polytopes approximating it. This is no longer the case for systems of qudits. For that reason, in the following, we concentrate on lower bounds of $\chi(\rho^{AB})$ by the inner polytope approximation for separability certification. Such inner polytope approximations can be used for an accurate and efficient certification of separability, even if the local dimensions are larger than two.

## 2.2 Adaptive polytopes

A key insight is that the structure of the procedure allows us to iteratively improve the choice of the inner polytope. To be precise, starting with a polytope $\mathcal{P}$ at Alice's side, one finds the set of unnormalised states $\{\tilde{\tau}_\lambda\}_{\lambda=1}^N$ at Bob's side in Eq. (5). Upon normalisation, these give an inner polytope $\mathcal{Q}$ to approximate the Bloch sphere at Bob's side, which can be used for the algorithm (5) with Alice and Bob being interchanged and after performing a system swap on the state $\rho$. Importantly, this newly obtained polytope forms an approximation that is at least as strong as the preceding polytope, which can be seen from the symmetrical roles of $\tilde{\tau}_\lambda$ and $\sigma_\lambda$ in the optimisation problem (5). Conceptually, it also makes no difference whether Alice's and Bob's system have the same or different dimensions. In the latter case, the size of the matrices $\tilde{\tau}_\lambda$ simply switches between every round of the algorithm. The polytope adaption algorithm may be summarised in the following way:

1. Initialise an arbitrary inner polytope at Alice's side $\mathcal{P}_A = \{\sigma_\lambda\}_{\lambda=1}^N$.

2. Compute $\chi_{\mathcal{P}_A}$ with respect to $\mathcal{P}_A$ by the SDP in Eq. (5), extract the corresponding $\{\tilde{\tau}_\lambda^B\}_{\lambda=1}^N$ and construct a polytope $\mathcal{P}_B$ with vertices $\{\tilde{\tau}_\lambda / \text{Tr}(\tilde{\tau}_\lambda)\}_{\lambda=1}^N$.

3. Exchange A and B, use $\mathcal{P}_B$ as polytope approximation and return to step 2.

The algorithm stops upon convergence of the visibility $\chi$.

These steps are illustrated in Fig. 2. This iterative procedure gives a series of SDPs approximating SEP with increasing accuracy, which in practice converges rather quickly. Also,

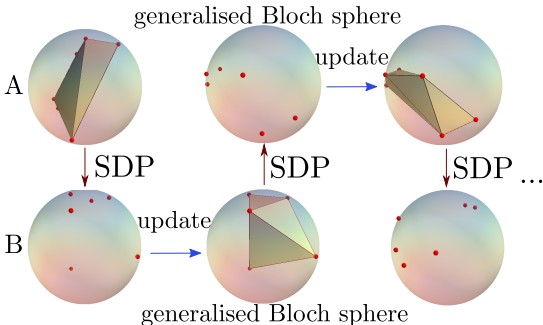

Figure 2: Schematic visualisation of polytope adaptions. The optimal feasible points computed in the SDP in Eq. (5) where Alice's generalized Bloch sphere is approximated by a polytope are used to construct a new polytope for the next approximation of Bob's generalized Bloch sphere. In this way, the algorithm of polytope adaptions alternates between polytope approximations of Alice's and Bob's system.

no significant difference in the performance is observed when using symmetric and random polytopes as initiation; in the following, the latter are used. It is also important to mention that the algorithm has a mild increase in complexity when number of iterations, polytope vertices and the Hilbert space dimension get higher. The number of scalar variables for a single SDP in one iteration step scales as $\mathcal{O}(Nd_B^2)$ for optimisation on Bob's side and as $\mathcal{O}(Nd_A^2)$ on Alice's side where $N$ is the number of polytope vertices and $d_B$ is the local dimension of Bob. Each iteration step has the same number of variables so that the runtime increases linearly in the number of iterations. Our implementation is written in the Julia programming language [33] and the semidefinite programs were solved using the Mosek solver [34]. The code is available on a public repository [35].

Our obtained inner approximation to SEP turns out to saturate various upper bounds such as those given by the positive partial transposition (PPT) criterion [36] or symmetric extensions [37] in all test cases, uncovering the optimality of both. For example, we obtain the exact values of $\chi$ for the isotropic and Werner states for local dimensions up to 10, and $10^4$ random states of dimension $5 \times 5$ distributed according to the Hilbert-Schmidt measure [38] with a numerical accuracy of $10^{-4}$. The computation time for states of a $5 \times 5$ system takes around $2-3$ seconds per iteration when the polytope has 200 vertices and 3 iterations are on average sufficient for obtaining the correct value. These times refer to a computer with a CPU Intel(R) Core(TM) i5-3570 processor (4 cores) running at 3.40GHz using 16 GB of RAM.

## 2.3 PPT-entangled states

We consider the two classic one-parameter families of quantum states of dimensions of $3 \times 3$ and $2 \times 4$ known as Horodecki states $\rho_{3 \times 3}^H(a)$ and $\rho_{2 \times 4}^H(b)$, where $0 \leq a, b \leq 1$ [39]; for the explicit density operators see Appendix E. Both are entangled despite being PPT for $0 < a, b < 1$ [39].

For the Horodecki state of dimension $3 \times 3$, many entanglement criteria have been used to obtain upper bounds for $\chi$, and certain lower bounds are also known. As seen in Fig 3a, our lower bound outperforms the best known lower bound [25] and approaches the best known upper bound given in Ref. [40]. For further comparison, we also implemented the SDP hierarchy of symmetric extensions to the fourth level for approximations of SEP from the outside [37].

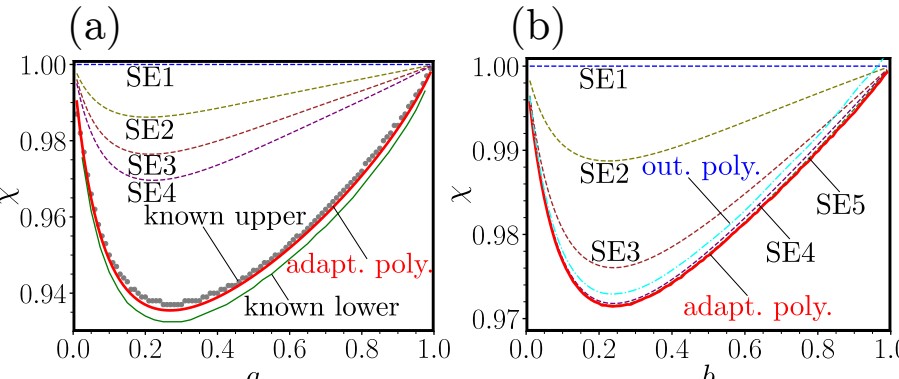

Figure 3: Plots of lower bound of the separability threshold for different state parameters of the $3 \times 3$ (a) and $2 \times 4$ (b) families of PPT-entangled states (red curves) obtained by polytope adaptions using 500 vertices. The dashed curves correspond to bounds given by different levels of symmetric extension (SE). The green and dotted curve in (a) labelled by "known lower" and "known upper" show the best previously known lower and upper bounds for the state family, see [25, 40].

In the case of $\rho_{2 \times 4}^{H}(b)$, the symmetric extension hierarchy is implemented up to level 5 on the qubit [37]. In Fig. 3b, we observe that the lower bound given by our inner adaptive polytope algorithm nearly coincides with this upper bound up to the numerical accuracy, convincingly demonstrating its optimality. Moreover, as Alice's Bloch sphere is a 3-dimensional unit sphere, the outer polytope approximation can also be easily constructed, albeit without adaptation. We choose a fixed outer polytope approximation with 1002 vertices, which already admits an entanglement detection generally better than symmetric extension of level 3, see Fig. 3b.

Importantly, the algorithm is applicable to generic states without special assumptions. This allows us to study and construct PPT entangled states beyond fine-tune families. Specifically, starting with a Horodecki PPT entangled state at $a = 0.25$ with visibility of $\chi[\rho_{2 \times 4}^{H}(0.25)] = 0.9715$, we design a see-saw algorithm to search for PPT states with higher entanglement robustness. In practice, we first compute the dual SDP of the polytope approximation (5) to obtain a witness for the corresponding approximation of SEP. In the second step, we maximise the violation of this witness over all PPT states. This new PPT entangled state is then chosen to be the input for the first SDP again. The see-saw algorithm stabilises at a state with significant lower visibility of 0.9461, which could be a candidate for an experimental realisation of robust bound entanglement. The method of polytope adaptions may therefore extend existing methods for the construction of bound entangled states [41].

## 3 Adaptive polytopes for multiparticle systems

### 3.1 Three-qubit systems

In multiparticle systems, one can distinguish between different types of entanglement and separability. Specifically, a state $\rho^{ABC}$ is called fully separable (FSEP) if it can be decomposed as convex combination of product states,

$$\rho^{ABC} = \sum_{\lambda} p_{\lambda}\, \sigma_{\lambda}^{A} \otimes \tau_{\lambda}^{B} \otimes \gamma_{\lambda}^{C}. \tag{6}$$

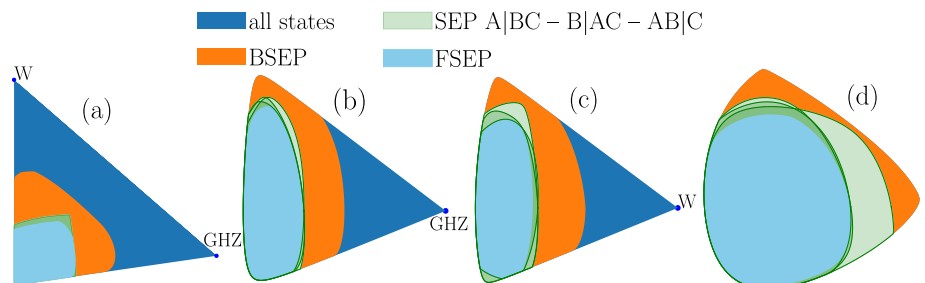

Figure 4: Two dimensional cross-sections of three-qubit states. The section represented in (a) corresponds to the plane that includes the GHZ-, W- and maximally mixed state. Note that the boundary for genuinely multiparticle entanglement is is already determined in Ref. [47]. In (b) and (c), the planes under consideration are obtained by the maximally mixed state, some random state and the GHZ and W state and in (d), two random states are taken into account.

Otherwise, it is entangled. The state $\rho^{ABC}$ is separable for the bipartition $A|BC$ if it can be written as

$$\rho^{ABC} = \sum_\lambda p_\lambda\, \sigma_\lambda^A \otimes \tau_\lambda^{BC}, \tag{7}$$

where $\tau_\lambda^{BC}$ may be entangled, and similarly for the remaining bipartitions $AB|C$ and $AC|B$. States that are separable for any bipartition are called fully biseparable (FBSEP). Finally, the set of biseparable (BSEP) quantum states are those which can be decomposed as a convex combination of states that are separable for these bipartions,

$$\rho^{ABC} = p_1\rho_1^{AB|C} + p_2\rho_2^{AC|B} + p_3\rho_3^{A|BC}, \tag{8}$$

where the superscripts indicate the membership to the corresponding separability class, e.g., $\rho_1^{AB|C} \in \text{SEP}(AB|C)$. Quantum states which are not biseparable are called genuinely multiparticle entangled.

To apply the adaptive polytope method for FSEP, we introduce a polytope $\mathcal{P}_A$ on the system $A$ and demand that the vertices of $\mathcal{P}_A$ are paired to positive-semidefinite operators on $BC$ that are PPT. This approach is valid as the set of PPT-states coincides with the separable states for two qubits.

Checking the membership to BSEP amounts to a choice of suitable polytopes $\mathcal{P}_A$, $\mathcal{P}_B$ and $\mathcal{P}_C$, one for each subsystem. Then, the feasible set in the SDP is given by the r.h.s. of Eq. (8) where each summand is replaced by the corresponding bipartite polytope approximation.

Similarly in the case of FBSEP, every subsystem is once approximated by a polytope but in opposition to BSEP it is demanded that the state $\rho^{ABC}$ is tested for membership to every bipartite separability class by a polytope approximation, see Appendix C.

As a demonstration, the inner approximation with a polytope of 300 vertices gives the white noise thresholds of 0.199 for full separability and 0.42857 for biseparability of the GHZ state in accordance with the known exact values [42,43]. In the case of the W state, we obtain a full separability threshold of 0.178 and 0.479 for biseparability which are both known to be exact [40,44]. Moreover, we are able to study certain two-dimensional cross-sections of the set of three qubit states, see Fig. 4. Remarkably, the introduced method delivers a suitable approximation for the set of fully separable states which is generally hard to achieve in contrast to the bipartite scenario. We use this advantage to study the set of fully biseparable states which are still entangled. Although examples for such states exist [45,46], the robustness of this phenomenon was unexplored.

| State | Lower bound | Upper bound |
|---|---|---|
| GHZ (5 Qubits) | 0.05878 | $1/17 \approx 0.05882^*$ [42] |
| W (5 Qubits) | 0.046956 | $0.0725^\sharp$ |
| Dicke (5 Qubits, 2 ex.) | 0.04226 | $0.05996^\sharp$ |
| GHZ (4 Qubits) | 0.1111 | $1/9 \approx 0.1111^*$ [42] |
| W (4 Qubits) | 0.0926 | $0.0926^*$ [48] |
| Cluster state (4 Qubits) | 0.111 | $0.111^\sharp$ |
| Dicke (4 Qubits, 2 ex.) | 0.08571 | $0.11111^\sharp$ |
| GHZ (3 Qutrits) | 0.0994 | 0.1 [49] |
| W (3 Qutrits) | 0.0602 | $0.0728^\sharp$ |

Table 1: Comparison of lower bounds for $\chi$ computed with the method of adaptive polytopes and known upper bounds from the literature. The three qutrit W, GHZ and four qubit Cluster states are defined by the vectors $\left|W_3^3\right\rangle = \frac{1}{\sqrt{6}}(|100\rangle + |010\rangle + |001\rangle + |200\rangle + |020\rangle + |002\rangle)$, $\left|\mathrm{GHZ}_3^3\right\rangle = \frac{1}{\sqrt{3}}(|000\rangle + |111\rangle + |222\rangle)$ and $\left|C_4\right\rangle = \frac{1}{2}(|{+}0{+}0\rangle + |{+}0{-}1\rangle + |{-}1{-}0\rangle + |{-}1{+}1\rangle)$. An asterisk ($*$) indicates the bound known to be tight. A sharp ($\sharp$) indicates the value obtained by the PPT criterion. The values always correspond to full separability. The computation time for the extreme cases (5 qubits, 3 qutrits) is a few minutes with the same hardware specification as mentioned above.

## 3.2 Robust fully biseparable states

We apply our method to approach the question on which fully biseparable state is maximally robust against full separability. To answer this question, first random entangled states are initialised and their visibility to FSEP is computed with the dual SDP from the polytope approximation. The solution provides an approximate entanglement witness, whose inner product with all fully bi-separable states is minimised in the second step. Then, the FSEP-visibility of the resulting state is computed and a corresponding new approximate entanglement witness is determined. These steps are repeated until convergence, see Appendix F. Although this iteration is not guaranteed to terminate in a global optimum, we obtain a stable minimal separability threshold $\chi \approx 0.57$. Further analysis shows that the corresponding obtained state admits a surprisingly simple form,

$$\rho(\theta) = \frac{1}{4} \sum_{\alpha,\beta=0}^{1} \left|\gamma_{\alpha\beta}(\theta)\right\rangle\left\langle\gamma_{\alpha\beta}(\theta)\right|, \tag{9}$$

for angles $\theta \in [0, 2\pi)$ that are solutions to $(\sin\theta\cos\theta)^2 = 1/6$, see Appendix F for a geometric analysis. The four vectors in expression (9) are

$$\left|\gamma_{\alpha\beta}\right\rangle = \frac{1}{\sqrt{2}}\left[i^\alpha \cos(\theta + \frac{\alpha\pi}{2})|0\rangle_A + (-1)^\beta \sin(\theta + \frac{\alpha\pi}{2})|1\rangle_A\right] \otimes \left|\psi_{\alpha\beta}\right\rangle_{BC}, \tag{10}$$

with

$$\left|\psi_{\alpha\beta}\right\rangle_{BC} = |\beta, \beta \oplus \alpha\rangle_{BC} - (-1)^\beta \left|\overline{\beta}, \overline{\beta \oplus \alpha}\right\rangle_{BC} \text{ for } \alpha, \beta \in \{0, 1\}, \tag{11}$$

where $\oplus$ denotes addition modulo 2 and $\overline{\alpha} = \alpha \oplus 1$, see Appendix F for more details.

## 3.3 More parties and higher dimensions

The idea of the adaptive algorithm can be further extended to certify full separability for systems of more parties and higher dimensions. Consider an $n$-partite state $\rho^{1\cdots n}$. One can initiate

a polytope approximation for the products states of $n-1$ parties, $\{\sigma_\lambda^1 \otimes \cdots \otimes \sigma_\lambda^{n-1}\}_{\lambda=1}^N$. One then can check whether there exist positive semidefinite matrices $\{\tilde\tau_\lambda^n\}_{\lambda=1}^N$ for the last party such that

$$\rho^{1\cdots n} = \sum_{\lambda=1}^N \sigma_\lambda^1 \otimes \cdots \otimes \sigma_\lambda^{n-1} \otimes \tilde\tau_\lambda^n. \tag{12}$$

This is again an SDP. The output of this SDP gives a polytope for the last party. One then iterates over the parties to systematically improve the approximation. In addition, for a pair of systems with total dimension lower than 6, one can use the PPT criterion to simplify the procedure. As an example, we study the full separability of up to five qubits as well as three qutrits. Comparison of the results with known states in the literature gives excellent agreement as indicated in Table 1. Extension to many-body quantum systems with more parties but with special structure or symmetry is interesting for future study.

## 4 Conclusion

We introduced a method to tackle the problem of certifying the separability of quantum states which is optimal under benchmarks. The resulting algorithm allows for a highly accurate description of the set of separable states from inside. The two main illustrative applications of the algorithm are the precise entanglement characterisation of states that cannot be found by the PPT criterion as well as a precise approximation of fully separable states in a multiparticle system of up to five qubits or three qutrits. Methodologically, we suggested a new approach to multi-linear optimisation problems that clearly demands for future applications. Concretely, related ideas can be useful in the study of quantum networks using local operations and shared randomness and in the characterisation of entanglement-breaking quantum channels. It is also directly applicable to the analysis of the role of memory in quantum processes building on the results in Ref. [50]. Future applications that might reach far beyond entanglement theory are also within the realm of possibility.

## Acknowledgements

The authors would like to thank Zhen-Peng Xu for inspiring discussions and comments and Jiangwei Shang for the supply of comparative data. The University of Siegen is kindly acknowledged for enabling our computations through the `OMNI` cluster.

**Funding information** This work was supported by the Deutsche Forschungsgemeinschaft (DFG, German Research Foundation, project numbers 447948357 and 440958198), the Sino-German Center for Research Promotion (Project M-0294), the ERC (Consolidator Grant 683107/TempoQ), and the German Ministry of Education and Research (Project QuKuK, BMBF Grant No. 16KIS1618K). X.D.Y. acknowledges support by the National Natural Science Foundation of China (Grants No. 12205170 and No. 12174224) and the Shandong Provincial Natural Science Foundation of China (Grant No. ZR2022QA084).

## A Semidefinite programming characterisation of $\mathrm{conv}(\mathcal{P} \otimes \mathcal{S}_B)$

In this appendix, we demonstrate how one can characterise $\mathrm{conv}(\mathcal{P} \otimes \mathcal{S}_B)$ for a polytope $\mathcal{P}$ by means of semidefinite programming. More specifically, let $\rho^{AB}$ be a bipartite state. We would

like to determine if $\rho^{AB} \in \text{conv}(\mathcal{P} \otimes \mathcal{S}_B)$. Quantitatively, we want to compute

$$\chi_{\mathcal{P}}(\rho^{AB}) = \max\{t \geq 0 : t\rho^{AB} + (1-t)\mathbb{1}/d_{AB} \in \text{conv}(\mathcal{P} \otimes \mathcal{S}_B)\}. \tag{13}$$

Let the polytope $\mathcal{P}$ be given by the vertices $\{\sigma_\lambda\}_{\lambda=1}^N$. The problem becomes

$$
\begin{aligned}
\chi_{\mathcal{P}}(\rho^{AB}) = \quad &\max \ t \\
&\text{w.r.t. } \{\tilde{\tau}_\lambda \geq 0\}, t \geq 0 \\
&\text{s.t. } t\rho^{AB} + (1-t)\mathbb{1}/d_{AB} = \sum_{\lambda=1}^N \sigma_\lambda \otimes \tilde{\tau}_\lambda.
\end{aligned}
\tag{14}
$$

For the sake of convenience, we formulate the dual problem for computing a related and equally useful quantity $\chi_{\mathcal{P}}(\rho^{AB}) = 1/[1 + r_{\mathcal{P}}(\rho^{AB})]$ with

$$
\begin{aligned}
-r_{\mathcal{P}}(\rho^{AB}) = \quad &\min \ \text{Tr}(\rho^{AB} W^{AB}) \\
&\text{w.r.t. } W^{AB} \\
&\text{s.t. } \text{Tr}(W^{AB}) = d_{AB} \\
&\qquad \text{Tr}_A(W^{AB}(\sigma_\lambda \otimes \mathbb{1})) \succcurlyeq 0 \ \forall \lambda.
\end{aligned}
\tag{15}
$$

The correspondence of (14) and (15) can be seen in two steps. In the first step, we derive an expression of the set of non-normalised separating hyperplanes for $\text{conv}(\mathcal{P} \otimes \mathcal{S}_B)$, which is the dual of the membership problem to $\text{conv}(\mathcal{P} \otimes \mathcal{S}_B)$. The membership problem is

$$
\begin{aligned}
&\max \quad 0 \\
&\text{w.r.t.} \quad \{\tilde{\tau}_\lambda \geq 0\} \\
&\text{s.t.} \quad \rho^{AB} = \sum_{\lambda=1}^N \sigma_\lambda \otimes \tilde{\tau}_\lambda,
\end{aligned}
\tag{16}
$$

and its dual reads

$$
\begin{aligned}
&\min \quad \text{Tr}(\rho^{AB} W^{AB}) \\
&\text{w.r.t.} \quad W^{AB} = (W^{AB})^\dagger \\
&\text{s.t.} \quad \text{Tr}_A(W^{AB}(\sigma_\lambda \otimes \mathbb{1})) \succcurlyeq 0 \ \forall \lambda.
\end{aligned}
\tag{17}
$$

This will then correspond to an outer approximation of the set of entanglement witnesses in the separability problem provided that $\mathcal{P}$ is an inner polytope.

In the second step, we use the representation of the entanglement robustness measure in terms of normalised entanglement witnesses derived in Ref. [31]. There, it is shown that the robustness with respect to white noise

$$r(\rho^{AB}) = \min\left\{s : \frac{\rho^{AB} + s\mathbb{1}/d_{AB}}{1+s} \in \text{SEP}\right\}, \tag{18}$$

can be equivalently expressed as

$$r(\rho^{AB}) = -\min_{Y^{AB} \in \mathcal{N}} \text{Tr}(Y^{AB}\rho^{AB}). \tag{19}$$

The set $\mathcal{N}$ on which the expression is minimised in (19) contains here entanglement witnesses with a special normalisation, namely

$$\mathcal{N} = \left\{Y^{AB} : Y^{AB} = (Y^{AB})^\dagger, \text{Tr}(Y^{AB}) = d_{AB}, \text{Tr}(Y^{AB}\rho^{AB}) \geq 0 \text{ for all } \rho^{AB} \in \text{SEP}\right\}. \tag{20}$$

In our case where we replace the set of states on $A$ by a polytope, we have to replace $\mathcal{N}$ in (19) by $\mathcal{N}_{\mathcal{P}}$, which is given by

$$\mathcal{N}_{\mathcal{P}} = \left\{Y^{AB} : Y^{AB} = (Y^{AB})^\dagger, \text{Tr}(Y^{AB}) = d_{AB}, \text{Tr}_A(Y^{AB}(\sigma_\lambda \otimes \mathbb{1})) \succcurlyeq 0 \ \forall \lambda\right\}, \tag{21}$$

to obtain $r_{\mathcal{P}}$. We can conclude that $\chi_{\mathcal{P}}(\rho^{AB}) = 1/[1 + r_{\mathcal{P}}(\rho^{AB})]$ with

$$r_{\mathcal{P}}(\rho^{AB}) = -\min_{Y^{AB} \in \mathcal{N}_{\mathcal{P}}} \text{Tr}(Y^{AB}\rho^{AB}), \tag{22}$$

which is in accordance to (15).

# B Estimation of other robustness measures

In our analysis, we focus on determining the visibility, or equivalently the robustness of entanglement with respect to white noise defined in Eq. (18). This is also often called random robustness [51]. In this appendix, we discuss other robustness measures of entanglement [51].

First, robustnesses resulting from different choices of separable states other than the maximally mixed state as noise mixing component in the random robustness (18) can be computed in a completely similar way. More generally, we can consider the so-called (absolute) robustness of entanglement with respect to the whole separable set, defined as [51]

$$R(\rho^{AB}) = \min_{\gamma^{AB} \in \text{SEP}} \min_{s \geq 0} \left\{ s : \frac{\rho^{AB} + s\gamma^{AB}}{1 + s} \in \text{SEP} \right\}. \tag{23}$$

Upon approximating Alice's set of states by a polytope $\mathcal{P}$ with vertices $\{\sigma_\lambda\}_{\lambda=1}^N$, $R(\rho^{AB})$ is upper bounded by $(1 - \bar{\chi}_\mathcal{P}(\rho^{AB}))/\bar{\chi}_\mathcal{P}(\rho^{AB})$, where

$$\bar{\chi}_\mathcal{P}(\rho^{AB}) = \max_{\gamma \in \text{conv}(\mathcal{P} \otimes \mathcal{S}_B)} \max_{t \geq 0} \left\{ t : t\rho^{AB} + (1-t)\gamma^{AB} \in \text{conv}(\mathcal{P} \otimes \mathcal{S}_B) \right\}. \tag{24}$$

This can be computed by the following semidefinite program:

$$\begin{aligned}
\bar{\chi}_\mathcal{P}(\rho^{AB}) = \quad &\max \quad t \\
&\text{w.r.t.} \quad \{\tilde{\tau}_\lambda \geq 0\}, \{\tilde{\eta}_\lambda \geq 0\}, t \geq 0 \\
&\text{s.t.} \quad t\rho^{AB} = \sum_{\lambda=1}^N \sigma_\lambda \otimes (\tilde{\tau}_\lambda - \tilde{\eta}_\lambda) \\
&\qquad\quad \sum_{\lambda=1}^N \text{Tr}(\tilde{\eta}_\lambda) = 1 - t.
\end{aligned} \tag{25}$$

In addition, the so-called generalised robustness, introduced in Ref. [52] as

$$R^G(\rho^{AB}) = \min_{\gamma^{AB} \in \mathcal{S}_{AB}} \min_{s \geq 0} \left\{ s : \frac{\rho^{AB} + s\gamma^{AB}}{1 + s} \in \text{SEP} \right\}, \tag{26}$$

can efficiently be computed with polytope approximations. The difference to the absolute robustness is that also entangled states are taken into account as possible sources of noise. Corresponding upper bounds of $R^G(\rho^{AB})$ achieved with a polytope with vertices $\{\sigma_\lambda\}_{\lambda=1}^N$ are given by $(1 - \chi_\mathcal{P}^G(\rho^{AB}))/\chi_\mathcal{P}^G(\rho^{AB})$ with

$$\begin{aligned}
\chi_\mathcal{P}^G(\rho^{AB}) = \quad &\max \quad t \\
&\text{w.r.t.} \quad \{\tilde{\tau}_\lambda \geq 0\}, \gamma^{AB} \geq 0, t \geq 0 \\
&\text{s.t.} \quad t\rho^{AB} + \gamma^{AB} = \sum_{\lambda=1}^N \sigma_\lambda \otimes \tilde{\tau}_\lambda \\
&\qquad\quad \text{Tr}(\gamma^{AB}) = 1 - t.
\end{aligned} \tag{27}$$

# C Polytope approximations in multiparticle systems

Turning to the multiparticle scenario, we start with considering the three-qubit system. We use $\alpha \in \{A, B, C\}$ to denote one of the three parties, and $\bar{\alpha}$ to denote its other complementary parties.

## C.1 Full separability

Mathematically, the set of fully separable states is defined by $\text{FSEP} = \text{conv}(\mathcal{S}_A \otimes \mathcal{S}_B \otimes \mathcal{S}_C)$. To construct the inner (outer) polytope approximation in this case, it is useful to observe that $\text{conv}(\mathcal{S}_A \otimes \mathcal{S}_B \otimes \mathcal{S}_C) = \text{conv}(\mathcal{S}_A \otimes \text{PPT}_{BC})$, where $\text{PPT}_{BC}$ denotes the set of two-qubit states

that have a positive partial transpose, which is the same as $\text{SEP}_{BC}$. By approximating $\mathcal{S}_A$ by a polytope $\mathcal{P} = \{\sigma_\lambda^A\}_{\lambda=1}^N$ from inside (outside), certification of a state $\rho^{ABC}$ to be fully separable amounts to finding $\{\tilde{\tau}_\lambda^{BC}\}_{\lambda=1}^N$ such that $\tilde{\tau}_\lambda^{BC} \geq 0$, $[\tilde{\tau}^{BC}]_\lambda^{T_B} \geq 0$ and

$$\rho^{ABC} = \sum_{\lambda=1}^N \sigma_\lambda^A \otimes \tilde{\tau}_\lambda^{BC}. \tag{28}$$

This is an SDP in which we test the membership to the convex set $\text{conv}(\mathcal{P} \otimes \text{PPT}_{BC})$. The primal and dual versions of the corresponding estimation of visibilities $\chi_{\mathcal{P}}^{\text{FSEP}}(\rho^{ABC})$ are given as follows:

| **Primal:** | **Dual:** | (29) |
|---|---|---|
| $\max t$ | $\min \text{Tr}(Y_0^{ABC}\rho^{ABC}) + 1$ | (30) |
| w.r.t. $t, \{\tilde{\tau}_\lambda^{BC}\}$ | w.r.t. $Y_0^{ABC}, \{Y_\lambda^{BC}\}$ | (31) |

$$\text{s.t. } t\rho^{ABC} + (1-t)\mathbb{1}/d_{ABC} = \sum_{\lambda=1}^N \sigma_\lambda^A \otimes \tilde{\tau}_\lambda^{BC} \qquad \text{s.t. } \text{Tr}\left[Y_0^{ABC}\left(\mathbb{1}/d_{ABC} - \rho^{ABC}\right)\right] = 1 \tag{32}$$

$$\tilde{\tau}_\lambda^{BC} \succcurlyeq 0 \ \forall\lambda \qquad\qquad \text{Tr}_A\left[(\sigma_\lambda^A \otimes \mathbb{1})Y_0^{ABC}\right] + Y_\lambda^{BC} \succcurlyeq 0 \ \forall\lambda \tag{33}$$

$$(\tilde{\tau}_\lambda^{BC})^{T_B} \succcurlyeq 0 \ \forall\lambda \qquad\qquad (-Y_\lambda^{BC})^{T_B} \succcurlyeq 0 \ \forall\lambda. \tag{34}$$

## C.2 Full biseparability

Motivated by separability of bipartite states, one can consider three-qubit states that are separable with respect to any bipartition. Formally, this set can be expressed as $\text{FBSEP} = \cap_\alpha \text{SEP}(\alpha|\bar{\alpha})$ (fully biseparable). Being able to systematically approximate $\text{SEP}(\alpha|\bar{\alpha})$ by the polytope approximation, we can also directly describe FBSEP.

In analogy to the already discussed cases of separability, FBSEP may be extended for general convex sets of operators $\mathcal{P}_\alpha = \{\sigma_\lambda^\alpha\}_{\lambda=1}^{N_\alpha}$ to $\text{FBSEP}(\mathcal{P}_A, \mathcal{P}_B, \mathcal{P}_C)$ defined by

$$\text{FBSEP}(\mathcal{P}_A, \mathcal{P}_B, \mathcal{P}_C) = \text{conv}(\mathcal{P}_A \otimes \mathcal{S}_{BC}) \cap \text{conv}(\mathcal{P}_B \otimes \mathcal{S}_{BC}) \cap \text{conv}(\mathcal{P}_C \otimes \mathcal{S}_{AB}). \tag{35}$$

The membership of $\rho^{ABC}$ to $\text{FBSEP}(\mathcal{P}_A, \mathcal{P}_B, \mathcal{P}_C)$ is then equivalent to the existence of positive-semidefinite operators $\{\tilde{\tau}_\lambda^{\bar{\alpha}}\}_{\lambda=1}^{N_\alpha}$ where $\tilde{\tau}_\lambda^{\bar{\alpha}}$ is an operator acting on $\bar{\alpha} = \{A, B, C\} \setminus \{\alpha\}$ such that

$$\sum_{\lambda=1}^{N_\alpha} \sigma_\lambda^\alpha \otimes \tilde{\tau}_\lambda^{\bar{\alpha}} \ \ \forall\alpha. \tag{36}$$

The corresponding semidefinite programm to compute the visibility of a state $\rho^{ABC}$ to FBSEP is given as

$$\chi_{\mathcal{P}_\alpha}^{FB}(\rho^{ABC}) := \max t \tag{37}$$

$$\text{w.r.t. } t, \{\tilde{\tau}_{\lambda_\alpha}^{\bar{\alpha}}\} \tag{38}$$

$$\text{s.t. } t\rho^{ABC} + (1-t)\mathbb{1}/d_{ABC} = \sum_{\lambda_C=1}^{N_C} \tilde{\tau}_{\lambda_C}^{AB} \otimes \sigma_{\lambda_C}^C \tag{39}$$

$$t\rho^{ABC} + (1-t)\mathbb{1}/d_{ABC} = \sum_{\lambda_A=1}^{N_A} \sigma_{\lambda_A}^A \otimes \tilde{\tau}_{\lambda_A}^{BC} \tag{40}$$

$$t\rho^{ABC} + (1-t)\mathbb{1}/d_{ABC} = \sum_{\lambda_B=1}^{N_B} \sigma_{\lambda_B}^B \otimes \tilde{\tau}_{\lambda_B}^{AC} \tag{41}$$

$$\tilde{\tau}_{\lambda_C}^{AB} \succcurlyeq 0, \tilde{\tau}_{\lambda_A}^{BC} \succcurlyeq 0, \tilde{\tau}_{\lambda_B}^{AC} \succcurlyeq 0 \text{ for all } \lambda_A, \lambda_B, \lambda_C. \tag{42}$$

### C.3  Biseparability

The weakest form of separability is defined as the convex hull of separable states with respect to all different bipartitions, $\text{BSEP} = \text{conv}(\cup_\alpha \text{SEP}(\alpha|\bar{\alpha}))$ where $\alpha \in \{A, B, C\}$ and $\bar{\alpha} = \{A, B, C\} \setminus \alpha$. Members of this set are called biseparable. Recalling that $\text{SEP}(\alpha|\bar{\alpha}) = \text{conv}(\mathcal{S}_\alpha \otimes \mathcal{S}_{\bar{\alpha}})$, let us now approximate the Bloch sphere $\mathcal{S}_\alpha$ by polytopes $\mathcal{P}_\alpha$ from inside (outside) for $\alpha = A, B, C$. Then one has $\text{conv}[\cup_\alpha \text{conv}(\mathcal{P}_\alpha \otimes \mathcal{S}_{\bar{\alpha}})]$ as an inner (outer) approximation for BSEP. One can also easily see that these inner (outer) approximations of BSEP can be described by SDPs. Fixing three inner (outer) polytope approximations of $N_\alpha$ vertices for the Bloch sphere, $\mathcal{P}_\alpha = \{\sigma_\lambda^\alpha\}_\lambda^{N_\alpha}$, certifying $\rho^{ABC} \in \text{conv}[\cup_\alpha \text{conv}(\mathcal{P}_\alpha \otimes \mathcal{S}_{\bar{\alpha}})]$ implies searching for three sets of operators $\{\tilde{\tau}_\lambda^{\bar{\alpha}}\}_{\lambda=1}^{N_\alpha}$, each respectively acting on the system $\bar{\alpha}$ with $\alpha = A, B, C$, such that

$$\rho^{ABC} = \sum_\alpha \sum_{\lambda=1}^{N_\alpha} \sigma_\lambda^\alpha \otimes \tilde{\tau}_\lambda^{\bar{\alpha}}, \tag{43}$$

which is an SDP. Explicitly, one can estimate the visibility of a state $\rho^{ABC}$ in the following way.

$$\chi_{\mathcal{P}_\alpha}^{\text{BSEP}}(\rho^{ABC}) := \max_{t, \{\tilde{\tau}_\lambda^{\bar{\alpha}}\}} t \tag{44}$$

$$\text{s.t. } t\rho^{ABC} + (1-t)\mathbb{1}/d_{ABC} = \sum_{\lambda=1}^{N_C} \tilde{\tau}_\lambda^{AB} \otimes \sigma_\lambda^C + \sum_{\lambda=1}^{N_A} \sigma_\lambda^A \otimes \tilde{\tau}_\lambda^{BC} + \sum_{\lambda=1}^{N_B} \sigma_\lambda^B \otimes \tilde{\tau}_\lambda^{AC} \tag{45}$$

$$\tilde{\tau}_\lambda^{AB} \succeq 0, \tilde{\tau}_\lambda^{BC} \succeq 0, \tilde{\tau}_\lambda^{AC} \succeq 0 \text{ for all } \lambda. \tag{46}$$

# D  Implementation of the polytope adaption technique

In this appendix, three different polytope adaption schemes for different situations of separability certification are explained. It will be discussed how the method can be implemented for the computation of the random robustness and absolute robustness in bipartite systems and the computation of the random robustness to full separability in multiparticle systems.

### D.1  Bipartite random robustness

In the case of bipartite systems and mixing with the maximally mixed state, both system parts are iteratively replaced by polytopes that are obtained by the feasible optimum of the preceding semidefinite program given by (14). The method of polytope adaptions may thus be summarised by the following simple algorithm:

1. Initialise an arbitrary inner polytope $\mathcal{P}_A$

2. Compute $\chi$ with respect to $\mathcal{P}_A$, extract the corresponding $\tilde{\tau}_i^B$ and construct a polytope $\mathcal{P}_B$ with vertices $\{\tilde{\tau}_i/\text{Tr}(\tilde{\tau}_i)\}$

3. Exchange A and B, i.e. update $\rho^{AB} \mapsto \Pi_{AB}\rho^{AB}\Pi_{AB}$ ($\Pi_{AB}$: permutes systems A and B) and $\mathcal{P}_B \mapsto \mathcal{P}_A$, and go back to step 2 if convergence is not achieved.

Figure 5 illustrates how the number of needed iterations depends on the number of initialised vertices.

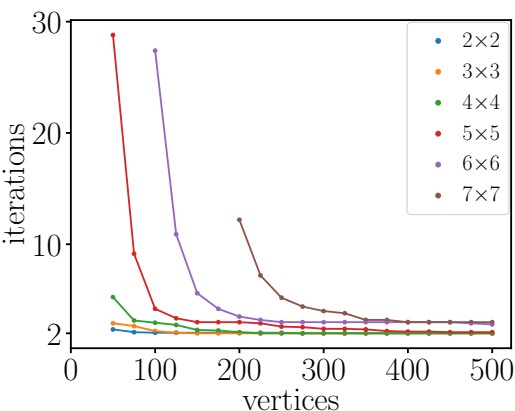

Figure 5: Comparison of the number of needed iterations for different numbers of polytope vertices using random states in $2 \times 2 - 7 \times 7$ dimensional systems. The algorithm stops if the difference of calculated values within one round of iteration is below $10^{-4}$. The convergence of the visibilities is confirmed by comparing with upper bounds given by the PPT criterion.

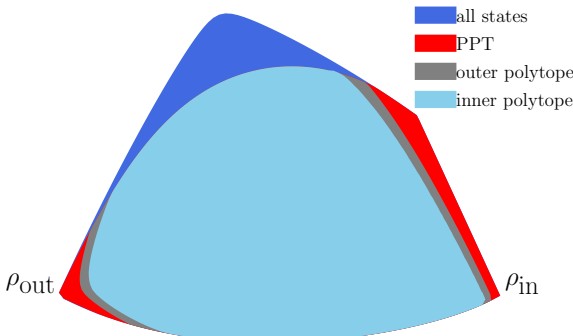

Figure 6: Cross-section of the 2-dimensional plane in the state space spanned by the most robust $2 \times 4$ Horodecki state $\rho_{\text{in}}$ and the obtained stronger entangled PPT state $\rho_{\text{out}}$. Remarkably, the stronger PPT entangled state is located in a different region of the global state space in the sense that their convex hull is mainly included in SEP.

### D.2 Bipartite absolute robustness

In the case in which one is interested in the robustness with respect to the whole separable set (see Section B), the method of polytope adaptions has to be slightly modified. Both of the lists of positive operators that one obtains as output from the first SDP (25) have to be used in the next iteration step. The list $\{\tilde{\eta}_\lambda^B\}$ must be invoked to construct a separable state and the list $\{\tilde{\tau}_\lambda^B\}$ for a new polytope approximation in every step of the iteration.

1. Initialise an arbitrary inner polytope $\mathcal{P}_A$ given by the vertices $\{\sigma_\lambda\}$

2. Compute $\bar{\chi}_{\mathcal{P}_A}$ with respect to $\mathcal{P}_A$ by the SDP (25), extract the corresponding $\tilde{\tau}_\lambda^B$ and $\tilde{\eta}_\lambda^B$ and define $\gamma^{AB} = \sum_\lambda \sigma_\lambda \otimes \tilde{\eta}_\lambda^B / \text{Tr}(\sum_\lambda \sigma_\lambda \otimes \tilde{\eta}_\lambda^B)$

3. Update $\mathcal{P}_B$ by $\{\tilde{\tau}_\lambda^B / \text{Tr}(\tilde{\tau}_\lambda^B)\}$, compute $\chi_{\mathcal{P}_B}^\gamma(\rho^{AB})$ given by the SDP

$$\max_{\{\sigma_\lambda\},t} \left\{ t \geq 0 : t\rho^{AB} + (1-t)\gamma^{AB} = \sum_\lambda \sigma_\lambda \otimes \tilde{\tau}_\lambda^B \right\}, \tag{47}$$

with $\gamma^{AB}$ extracted from the previous SDP.

4. Evaluate the corresponding $\sigma_\lambda$, update $\mathcal{P}_A$ with the vertices $\{\sigma_\lambda/\mathrm{Tr}(\sigma_\lambda)\}$ and go back to step 2 if convergence is not reached.

### D.3 Full separability in multiparticle systems

One situation in which polytope adaptions can also be applied is the treatment of full separability of a multiparticle quantum state. As described in the main text, we applied the method for states consisting of up to five qubits or up to three qutrits. We use a cyclic optimisation over all local subsystems. That means, in each iteration step all but one chosen local subsystem are approximated by a polytope. The feasible solution of this SDP in one iteration step then determines a new polytope for this chosen local subsystem. In the next iteration, another local system is chosen. In the case of systems involving qubits and qutrits, it is also possible optimise over two subsystems in one iteration by making use of the PPT criterion when it is necessary and sufficient for separability.

## E Robust PPT-entangled states

The method of polytope approximations may be used to construct robust PPT-entangled states. To do this, one has to initialise some PPT entangled state $\rho_{\mathrm{in}}^{AB}$. A possible choice of such initial states are members of the families of Horodecki states $\rho_{3|3}^H(a)$ and $\rho_{2|4}^H(b)$ given by

$$
\rho_{3|3}^H(a) = \frac{1}{8a+1}
\begin{pmatrix}
a & 0 & 0 & 0 & a & 0 & 0 & 0 & a \\
0 & a & 0 & 0 & 0 & 0 & 0 & 0 & 0 \\
0 & 0 & a & 0 & 0 & 0 & 0 & 0 & 0 \\
0 & 0 & 0 & a & 0 & 0 & 0 & 0 & 0 \\
a & 0 & 0 & 0 & a & 0 & 0 & 0 & a \\
0 & 0 & 0 & 0 & 0 & a & 0 & 0 & 0 \\
0 & 0 & 0 & 0 & 0 & 0 & \frac{1+a}{2} & 0 & \frac{\sqrt{1-a^2}}{2} \\
0 & 0 & 0 & 0 & 0 & 0 & 0 & a & 0 \\
a & 0 & 0 & 0 & a & 0 & \frac{\sqrt{1-a^2}}{2} & 0 & \frac{1+a}{2}
\end{pmatrix}
\tag{48}
$$

$$
\rho_{2|4}^H(b) = \frac{1}{7b+1}
\begin{pmatrix}
b & 0 & 0 & 0 & 0 & b & 0 & 0 \\
0 & b & 0 & 0 & 0 & 0 & b & 0 \\
0 & 0 & b & 0 & 0 & 0 & 0 & b \\
0 & 0 & 0 & b & 0 & 0 & 0 & 0 \\
0 & 0 & 0 & 0 & \frac{1+b}{2} & 0 & 0 & \frac{\sqrt{1-b^2}}{2} \\
b & 0 & 0 & 0 & 0 & b & 0 & 0 \\
0 & b & 0 & 0 & 0 & 0 & b & 0 \\
0 & 0 & b & 0 & \frac{\sqrt{1-b^2}}{2} & 0 & 0 & \frac{1+b}{2}
\end{pmatrix} .
\tag{49}
$$

A reasonable choice of such a state is for example the $2 \times 4$ dimensional Horodecki state at its most entangled position, i.e. at $b = 0.25$. The strategy to construct robust PPT-entangled states involves then the following steps:

1. Set $t = 0$.

2. Perform a polytope adaption for $\rho_{\mathrm{in}}^{AB}$, set $t = \chi(\rho_{\mathrm{in}}^{AB})$ and save the resulting polytope $\mathcal{P}_{\mathrm{res}}$ on Alice's side. If the value of $t$ did not change, exit.

3. Perform the dual SDP (15) with respect to $\mathcal{P}_{\mathrm{res}}$ and save the resulting approximate entanglement witness $Y^{AB}$.

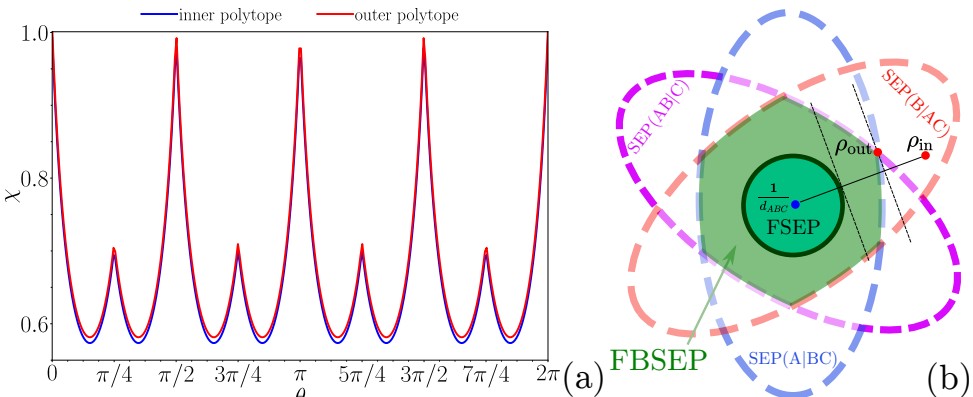

Figure 7: (a): Plot of the maximal mixing parameter of $\rho(\theta)$ for different angles $\theta$. Minimas of the curve correspond to points that are found by the optimisation procedure. (b): A sketch of the algorithm to find local maximally robust fully biseparable states. Approximate witnesses to the set FSEP are constructed followed by a minimisation of the inner product of this operator among all fully biseparable states.

4. Minimise the overlap $\mathrm{Tr}(Y^{AB}\rho^{AB})$ over all states $\rho^{AB}$ that have a positive partial transpose to obtain $\rho_{\mathrm{out}}^{AB}$.

5. Update $\rho_{\mathrm{in}}^{AB} = \rho_{\mathrm{out}}^{AB}$ and go back to step 2.

The resulting robust PPT-entangled state $\rho_{\mathrm{res}}^{AB}$ that we obtain has a visibility of $\chi(\rho_{\mathrm{res}}^{AB}) = 0.9461$. In Fig. 6, the cross section spanned by the initial and the resulting states is plotted. It can be seen that the algorithm ended up in an unrelated PPT-entangled state lying in a different corner of the set of states.

## F Robust fully biseparable states

To find and optimise states that are separable with respect to any bipartition but still not fully separable we invoke a see-saw algorithm which is very similar to the one used for robust PPT-entangled states. We followed precisely the following steps to collect states which are locally maximally robust:

1. Set $t = 0$ and initialise a random three-qubit state $\rho_{\mathrm{in}}^{ABC}$.

2. Perform a polytope adaption for $\rho_{\mathrm{in}}^{ABC}$, set $t = \chi(\rho_{\mathrm{in}}^{ABC})$ and save the resulting polytope $\mathcal{P}_{\mathrm{res}}$ on Alice's side. If the value of $t$ did not change, exit.

3. Perform the dual SDP (30) – (34) with respect to $\mathcal{P}_{\mathrm{res}}$ and save the resulting approximate entanglement witness $Y^{ABC}$.

4. Minimize the overlap $\mathrm{Tr}(Y^{ABC}\rho^{ABC})$ over all states $\rho^{ABC}$ that are fully biseparable to obtain $\rho_{\mathrm{out}}^{ABC}$. If the value is non-negative, go back to step 1 (thresholds for fully separable and fully biseparable coincide).

5. Update $\rho_{\mathrm{in}}^{ABC} = \rho_{\mathrm{out}}^{ABC}$ and go back to step 2.

A state which results from this algorithm may be further transformed with local unitary operators for the minimisation of matrix entries which helps in the analysis of that state. The states

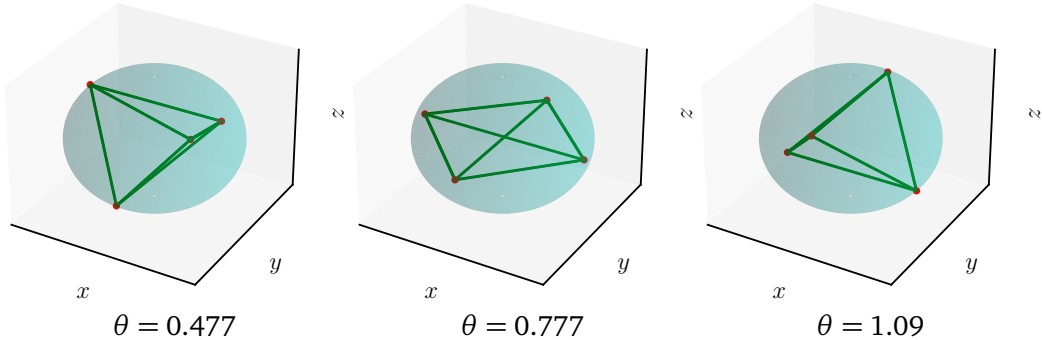

$\theta = 0.477$ $\qquad\qquad$ $\theta = 0.777$ $\qquad\qquad$ $\theta = 1.09$

Figure 8: Reduced states $\mathrm{Tr}_{BC}(|\gamma_i(\theta)\rangle\langle\gamma_i(\theta)|)$ (red points) and the corresponding connection lines (green lines) for different angles. From left to right $\theta = (0.477, 0.777, 1.09)$.

that we obtain with this procedure take the form

$$\rho(\theta) = \frac{1}{4}\sum_{i=1}^{4}|\gamma_i(\theta)\rangle\langle\gamma_i(\theta)|\,, \tag{50}$$

spanned by the vectors

$$|\gamma_1(\theta)\rangle = \frac{1}{\sqrt{2}}(\cos\theta\,|0\rangle_A + \sin\theta\,|1\rangle_A)\otimes(|00\rangle_{BC} - |11\rangle_{BC}) \tag{51}$$

$$|\gamma_2(\theta)\rangle = \frac{1}{\sqrt{2}}(-i\sin\theta\,|0\rangle_A + \cos\theta\,|1\rangle_A)\otimes(|10\rangle_{BC} - |01\rangle_{BC}) \tag{52}$$

$$|\gamma_3(\theta)\rangle = \frac{1}{\sqrt{2}}(-\cos\theta\,|0\rangle_A + \sin\theta\,|1\rangle_A)\otimes(|00\rangle_{BC} + |11\rangle_{BC}) \tag{53}$$

$$\left|\gamma_4(\theta)\right\rangle = \frac{1}{\sqrt{2}}(-i\sin\theta\,|0\rangle_A - \cos\theta\,|1\rangle_A)\otimes(|10\rangle_{BC} + |01\rangle_{BC})\,. \tag{54}$$

for certain angles $\theta$. Notice that the index $i$ here corresponds to $(\alpha,\beta)$ in the main text. Hence, each member in the family of states $\rho(\theta)$ is proportional to a projection on a four-dimensional subspace that is spanned by A$-$BC product vectors. For them, we have one of the Bell states on the BC$-$system and some superposition parametrised by a coherence angle $\theta$ on the A$-$system, respectively. The state family shares some noticeable similarity with the post measurement state in the teleportation protocol using Bell-states [53]. The only difference between them is the appearance of a factor $i$ in $|\gamma_2(\theta)\rangle$ and $\left|\gamma_4(\theta)\right\rangle$. Physically, this difference is expressed by an extra phase flip operation on the subsystem A conditioned on measuring the Bell-states $|\psi^+\rangle$ or $|\psi^-\rangle$. A comparison with the state without the respective phase flips reveals that the robustness is drastically increased. To see that $\rho(\theta)$ is fully biseparable for all $\theta$, it is useful to note that $\rho(\theta)$ is "X-shaped" in the computational basis, i.e. we have

$$\rho(\theta) = \frac{1}{4}\begin{pmatrix} a & 0 & 0 & 0 & 0 & 0 & 0 & -c \\ 0 & b & 0 & 0 & 0 & 0 & ic & 0 \\ 0 & 0 & b & 0 & 0 & ic & 0 & 0 \\ 0 & 0 & 0 & a & -c & 0 & 0 & 0 \\ 0 & 0 & 0 & -c & b & 0 & 0 & 0 \\ 0 & 0 & -ic & 0 & 0 & a & 0 & 0 \\ 0 & -ic & 0 & 0 & 0 & 0 & a & 0 \\ -c & 0 & 0 & 0 & 0 & 0 & 0 & b \end{pmatrix}\,, \tag{55}$$

with $a = \cos^2(\theta)$, $b = \sin^2(\theta)$ and $c = \sin(\theta)\cos(\theta)$. By the necessary and sufficient criteria on bipartite separability of three qubit X-shaped states given in [54], it follows

that $\rho(\theta)$ is fully biseparable according to Eqns. (1) - (3) in Ref. [54] if and only if $\sqrt{\cos^2(\theta)\sin^2(\theta)} \geq |\sin(\theta)\cos(\theta)|$ which is trivially fulfilled by equality for all $\theta$.

On the other hand, it is still an interesting question to ask for which angles the state $\rho(\theta)$ is most robust against full separability when mixing with white noise. There are eight points between 0 and $2\pi$ for which the visibility is minimal, see Fig. 7, and they are given by $\theta_{1-4} = \approx 0.48 + n\pi/2$ and $\theta_{5-8} \approx 1.09 + n\pi/2$ ($n = 0, 1, 2, 3$). A reason for this exact numerical values may be seen when considering the reduced states of $|\gamma_i(\theta)\rangle$ on the A subsystem. Their Bloch vectors take the form

$$
\begin{aligned}
r_1 &= \begin{pmatrix} 2\cos(\theta)\sin(\theta) \\ 0 \\ \cos^2(\theta)-\sin^2(\theta) \end{pmatrix}, & r_2 &= \begin{pmatrix} 0 \\ -2\cos(\theta)\sin(\theta) \\ \sin^2(\theta)-\cos^2(\theta) \end{pmatrix}, \\
r_3 &= \begin{pmatrix} -2\cos(\theta)\sin(\theta) \\ 0 \\ \cos^2(\theta)-\sin^2(\theta) \end{pmatrix}, & r_4 &= \begin{pmatrix} 0 \\ 2\cos(\theta)\sin(\theta) \\ \sin^2(\theta)-\cos^2(\theta) \end{pmatrix},
\end{aligned}
\tag{56}
$$

so that their mutual differences are either given by $||r_1 - r_2||^2 = 4(\cos^4(\theta) + \sin^4(\theta))$ or $||r_1 - r_3||^2 = 16(\cos^2(\theta)\sin^2(\theta))$. A straightforward calculation reveals that all vectors have the same distance (and therefore span a regular tetrahedron inside the Bloch sphere, see Fig. 8), exactly when the equation $\cos^2(\theta)\sin^2(\theta) = 1/6$ is fulfilled, e.g. for $\theta = \arccos\left(\sqrt{\frac{1}{2} + \frac{1}{\sqrt{12}}}\right)$. Interestingly those are exactly the angles $\theta_{1-8}$ for which the robustness of $\rho(\theta)$ is maximised. The local maxima at e.g. $\theta \approx 0.777$ correspond to the situation in which the Bloch vectors lie in a two-dimensional plane, see Fig. 8.

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
