# Peer review of "Certifying Quantum Separability with Adaptive Polytopes"

_SciPost Physics, doi:SciPost Phys. 16, 063 (2024)_

## Round 2 · Referee Report · Anonymous (Referee 1) · 2023-8-17

Report

The present manuscript describes an approximate method to determine nonseparability of a given state. This is done by approximating local sets of states by polytopes. Subsequently, it is tested whrether the state does belong into the convex hull of the product of such polytopes.
I think the manuscrip is presented in a rather clear manner, but there are a few points that definetely need an improvement. For example, the description of the algorithm of the adaptive plytopes is unclear. The question is what is changed between the rounds. Is it only the direction of the $tau$s, or is it also their number. Figure 2 is not really informative in this respect, it just shows switching sides. The next question is how exactly the state is determined to be in or outside the large polytope, I expect it to be by the wonder of semidefinite programming, but if there more details, I would be happy to learn them and the Readers with me. After all, this is the whole point of this game. Also, does tis procedure has to be repeated for every new state tested, or it constitues a "database" for future states. Then, what is the actual computational complexity of the protocol and how does it scale (roughly) with . Another point is that Figure 4 utilizes mixtures of random, unknown states, so their information content is very limited. Finally, the review on recent advances on entanglement detection is rather selective and not quite close to complete. The Authors should update the literature. At this moment, I am weakly not in favour of accepting the manuscript to SciPost Physics, but a revision is definetely needed and i would rather postpone my recommendation until then
  • validity: high
  • significance: good
  • originality: good
  • clarity: ok
  • formatting: good
  • grammar: perfect

Author:  Ties-Albrecht Ohst  on 2023-10-26  [id 4067]

(in reply to Report 1 on 2023-08-17)

Thank you for your effort in reviewing our manuscript. We have revised the manuscript taking your comments into account. Below we discuss the points you have raised.

**The Reviewer wrote:**
>For example, the description of the algorithm of the adaptive plytopes is unclear. The question is what is changed between the rounds. Is it only the direction of the taus, or is it also their number. Figure 2 is not really informative in this respect, it just shows switching sides.

**Our response:**

In consideration that the algorithm might not be as clear as we intended to, we have now added several paragraphs in Section 2.2.
In addition, the caption Figure 2 is extended to accommodate more information about the algorithm.
We hope that with this, the algorithm becomes clearer to the readers.

In short, to clarify your question, the party whose set of states is replaced by a polytope is changed in every round of the iteration.
We also would like to keep Figure 2, which should become clearer with the new text added.

**The Reviewer wrote:**
>The next question is how exactly the state is determined to be in or outside the large polytope, I expect it to be by the wonder of semidefinite programming, but if there more details, I would be happy to learn them and the Readers with me. After all, this is the whole point of this game. Also, does tis procedure has to be repeated for every new state tested, or it constitues a "database" for future states. Then, what is the actual computational complexity of the protocol and how does it scale (roughly) with.

**Our response:**

We have the impression that there might be a misunderstanding, that the polytopes used in our algorithm approximate the whole set of separable states.
This is not the case.
It is important to emphasize that the polytopes used in our algorithm are intended to approximate one of the local state spaces, either the one of Alice or Bob.
They are not an approximation for the global set of separable states shared by Alice and Bob.
The generated approximation of the global set of separable states is not a polytope.

Regarding the question whether the procedure has to be repeated for every new input state, the answer is `yes'. As long as there is no knowledge what the separable decomposition of a state could be, initialising a new random polytope is a fairly good option. However, this has no significant effects on the complexity of the algorithm.

The complexity of the algorithm scales linearly in the size of the polytope and quadratically in the local dimension on one side. A paragraph in the Section 2.2. is added to explain this aspect.

**The Reviewer wrote:**
>Another point is that Figure 4 utilizes mixtures of random, unknown states, so their information content is very limited.

**Our response:**

The purpose of Figure 4 is to emphasize on the generic nature of the algorithm, which can be applied to a general quantum state without any special structure. Once the state has more structure or symmetry (such as a GHZ state or the W state alone), determining their separability could be much more easier as known in the literature. With that in mind, we still want to keep this figure in this resubmission.

**The Reviewer wrote:**
>Finally, the review on recent advances on entanglement detection is rather selective and not quite close to complete. The Authors should update the literature.

**Our response:**

Notice that we are concentrating on the problem of verification of separability. This is complementary to the problem of entanglement detection. As we emphasized in the introduction, there are many methods for entanglement detection and the corresponding literature is indeed large. The verification of separability has been long considered to be a difficult problem, for which there are only handful of methods, which are mentioned in the introduction to our best knowledge.

However, we do agree that adding more references on entanglement detection would put the work into a larger context. We have added five recent works to our references (see Ref.[9-13]) regarding entanglement detection in this resubmission.

**The Reviewer wrote:**
>At this moment, I am weakly not in favour of accepting the manuscript to SciPost Physics, but a revision is definetely needed and i would rather postpone my recommendation until then.

**Our response:**
With our clarification, we hope that it would meet the standard of SciPost Physics.

---

## Round 2 · Referee Report · Anonymous (Referee 2) · 2023-10-3

Report

The manuscript addresses the significant issue of characterizing entanglement and separability in quantum states, a topic of great relevance in various fields of physics. The authors introduce a novel method based on adaptive polytope approximation to certify quantum separability in two- and multiparticle quantum systems. Their approach presents a practical algorithm that can conclusively identify two-particle separability for small and medium-sized dimensions and characterize full separability for multiparticle systems, such as up to five qubits or three qutrits. Furthermore, it allows for the distinction of different classes of entanglement and the identification of quantum states with intriguing entanglement properties, including maximally robust states.

The manuscript presents a valuable contribution to the field of quantum entanglement and separability certification, addressing a gap left by previous computationally demanding methods. To me, it seems they are using See-saw type coupled SDPs. The proposed algorithm demonstrates efficiency and versatility, making it applicable to quantum systems with relatively high dimensions and many particles. It offers the ability to certify a range of states and investigate various entanglement robustnesses. I recommend accepting the manuscript in its current form, as the presented approach is straightforward, useful, and appears to be highly effective.
  • validity: -
  • significance: -
  • originality: -
  • clarity: -
  • formatting: -
  • grammar: -

Author:  Ties-Albrecht Ohst  on 2023-10-26  [id 4066]

(in reply to Report 2 on 2023-10-03)

Thank you for your effort in reviewing our manuscript.

We do hope that our results contribute to the better understanding of the problem of certifying separability in entanglement theory.

---

## Round 3 · Referee Report · Anonymous (Referee 1) · 2023-11-14

Strengths

The paper presents an algorithm intended to certify separability of a given state. The algorimth seems to be very efficient computationally, but this needs more clarifications. Certainly, the Authors achieve rather precise estimates.

Weaknesses

Some details still require clarification. In particular I expect improvement on describing the assymmetry of the system. With the new version I am somewhat confused if the algorithm also certifies entanglement.

Report

I thank the Authors for replying to my previous report. However, I am not sure if I can find the amendmments satisfactory enough at this stage. The Authors claim that the Algorithm certifies separability and such a message can be read from the description of the algorithm. Moreover, Figure 5 suggests that convergence may be expected even for 2 iterations. However, I, and I believve the Reader, would like to know about the omplexity of the algorithm. In each step, a semi-definite program must be run, and we do not know much about optimizing $tau$ operators. I think it would be beneficial to give runtimes for a few examples discusssed in the paper in context of the specs of a computer used.
Another question that would need to be discussed is the symmetry under the swap of particles. The algorithm is presented as for a system of two identical subsystems. Shall the Reader assume that in case of, say, 2x4 we simply expand the space to 4x4? But what about taus being now sigmas?
In the newest version, the Author now claim that calculating both P_in and P_out is efficient. One of the versions of the dual problem we optimize Tr rho W, suggesting an entanglement witness. Table 5 quotes both upper and lower bounds of white noise robustness. Thus I am surprised that the Authors clarified in their response that they certify separability. The above hints me towards believing that the Algorithm is indeed able to provide an entanglement witness. I ask the Authors to confirm or deny. Also, If witnesses are provided, then despite the recent literature was updated, it still lack some references relevant to finding custom-tailored entanglement witnesses. For example, it was Brandao to first notice the relation between an entanglement witness and a separble approximation of a state.
Finally, as a remark, I would point out that the stop criterion in the caption of Figure 5 is not informative, as it ignores the dynamics of the subsequnt iterations. If many rounds of the procedure offer the correction of 10^-4 each, we are still far away. On the other hand, a lot is known about robustness of two-qudit Werner states, and it is difficult to infer how close the Algorithm has got. Only from other data it seems that it is rapidly convergent.
I am still in favour of publishing the work in SciPost, but basing on the above comments, I think there is still some space to for improvements to make it more useful to the Reader.

Requested changes

  1. Explain the computational cost of the algorithm and give runtimes for few examples discussed in the paper.
  2. Explain the procedure under asmymetric systems and states.
  3. Explain if the newest version applies to both the inner and the outer polytope, providing both the upper and the lower limits of robustness.
  4. If the answer to the last question is "yes", the literature on custom-tailored entanglement witnesses can be updated.

  • validity: high
  • significance: good
  • originality: high
  • clarity: good
  • formatting: excellent
  • grammar: perfect

Author:  Ties-Albrecht Ohst  on 2024-01-10  [id 4235]

(in reply to Report 1 on 2023-11-14)

Thank you for your effort in reviewing our manuscript. We have revised the manuscript taking your comments into account. Below we discuss the points you have raised.

**The Reviewer wrote:**

I thank the Authors for replying to my previous report. However, I am not sure if I can find the amendmments satisfactory enough at this stage. The Authors claim that the Algorithm certifies separability and such a message can be read from the description of the algorithm. Moreover, Figure 5 suggests that convergence may be expected even for 2 iterations. However, I, and I believve the Reader, would like to know about the omplexity of the algorithm. In each step, a semi-definite program must be run, and we do not know much about optimizing tau operators. I think it would be beneficial to give runtimes for a few examples discusssed in the paper in context of the specs of a computer used.

**Our response:**

Thank you for your comment. We agree, and now explicitly give some examples of run times for certain states so that the readers have an idea of how long it typically takes. This is given at the end of section 2.2 for random 5x5-dimensional states (order of seconds in a normal desktop) and in the Caption of Table 1 for multiparticle states (order of minutes in a normal desktop).

**The Reviewer wrote:**

Another question that would need to be discussed is the symmetry under the swap of particles. The algorithm is presented as for a system of two identical subsystems. Shall the Reader assume that in case of, say, 2x4 we simply expand the space to 4x4? But what about taus being now sigmas?

**Our response:**

No, the spaces do not have to be expanded. Our algorithm switches between optimisations on Alice and Bob's generalized Bloch spheres which may have different dimensions. Importantly, in the third step of the algorithm, in which the systems A and B are exchanged, a swap operation on the state is performed which transforms a 2x4-dimensional state into an 4x2-dimensional state. Hence, every odd iteration step involves an optimisation of 2x2 matrices, while every even iteration step involves an optimisation of 4x4 matrices (or the other way around). In this way, the algorithm does work normally in the asymmetric setting without any additional increase in complexity compared to the symmetric setting.
In the revision, we clarified this in section 2.2.

**The Reviewer wrote:**

In the newest version, the Author now claim that calculating both P_in and P_out is efficient. One of the versions of the dual problem we optimize Tr rho W, suggesting an entanglement witness. Table 5 quotes both upper and lower bounds of white noise robustness. Thus I am surprised that the Authors clarified in their response that they certify separability. The above hints me towards believing that the Algorithm is indeed able to provide an entanglement witness. I ask the Authors to confirm or deny. Also, If witnesses are provided, then despite the recent literature was updated, it still lack some references relevant to finding custom-tailored entanglement witnesses. For example, it was Brandao to first notice the relation between an entanglement witness and a separble approximation of a state.

**Our response:**

Sorry for the confusion. Inner polytopes provide an inner approximation of the separable states whereas outer polytopes give rise to an outer approximation. Hence, entanglement detection is indeed possible by using the outer polytope approximation. However, the outer polytope approximation is only practical if one of the systems is a qubit, since then its low-dimensional Bloch sphere allows for an efficient construction of an outer polytope approximation to it. Such a construction is unknown for qudits. Also, the adaption techniques we developed for the inner polytopes cannot be applied to the outer approximation. We added few sentences in the end of Section 2.1. together with references on customized entanglement witnesses to clarify this. On the other hand, the algorithm with adaptive inner polytopes allows for certification of separability also for system of qudits. As this problem has been long considered to be difficult, we mainly concentrate on this aspect in the rest of the manuscript.

**The Reviewer wrote:**

Finally, as a remark, I would point out that the stop criterion in the caption of Figure 5 is not informative, as it ignores the dynamics of the subsequnt iterations. If many rounds of the procedure offer the correction of 10^-4 each, we are still far away. On the other hand, a lot is known about robustness of two-qudit Werner states, and it is difficult to infer how close the Algorithm has got. Only from other data it seems that it is rapidly convergent.

**Our response:**

Thank you for pointing this out. The convergence of the Figure 5 was in fact confirmed by comparing with upper bounds given by the PPT criterion (which is good for random states as we discussed in section 2.2.) We revise the caption to make this clear.

**The Reviewer wrote:**

I am still in favour of publishing the work in SciPost, but basing on the above comments, I think there is still some space to for improvements to make it more useful to the Reader.

**Our response:**

Thank you, your comments have indeed improved the clarity of the manuscript!

---

## Round 3 · Author Response

This resubmission has been made in full consideration of the reports from the two Reviewers.

Specific replies to the comments from the Reviwers were also made.

---

## Round 3 · List of Changes

The revision includes:

-- changed to the SciPost LaTeX template

-- included more references on entanglement detection in the introduction

-- more detailed description of the polytope adaption algorithm (Ch. 2.2).

-- added short description of the complexity of the algorithm (Ch. 2.2).

-- extension of the caption of Figure 2

-- merged and restructured Figure 7 and 8 in Appendix F.

---

## Round 4 · Author Response

This minor revision has been made in consideration of the comments and requested changes from the report. A specific reply to the comments from the Reviewer has also been made.

---

## Round 4 · List of Changes

The revision includes:

-- added a clarification of the strengths and weaknesses of outer polytopes in the end of section 2.1

-- added information about the algorithm for systems with asymmetric dimensions in section 2.2

-- specified computation time for examples with hardware and software information in section 2.2 and in the caption of Table 1

-- added references to Section 2.1 and 2.2

-- clarified the used convergence criterion in the caption of Figure 5

---

## Editorial Decision

published